# Endothelial Dysfunction Is Associated with Cerebrovascular Events in Pre-Dialysis CKD Patients: A Prospective Study

**DOI:** 10.3390/life11020128

**Published:** 2021-02-07

**Authors:** Ana Cerqueira, Janete Quelhas-Santos, Susana Sampaio, Inês Ferreira, Miguel Relvas, Nídia Marques, Cláudia Camila Dias, Manuel Pestana

**Affiliations:** 1Nephrology Department, Centro Hospitalar Universitário São João, 4200-319 Porto, Portugal; susana.sampaio@sapo.pt (S.S.); inescastroferreira@sapo.pt (I.F.); mic21892@gmail.com (M.R.); nidiaa_92@hotmail.com (N.M.); mjfpestana@gmail.com (M.P.); 2Department of Medicine, Faculty of Medicine, University of Porto, 4200-250 Porto, Portugal; 3Institute for Innovation and Health Research (I3S), Institute of Biomedical Engineering (INEB), Nephrology and Infectious Diseases Research Group, University of Porto, 4200-250 Porto, Portugal; 4Faculty of Medicine, University of Porto, 4200-250 Porto, Portugal; sjanete@med.up.pt; 5Department of Community Medicine Health Information and Decision, Faculty of Medicine, University of Porto, 4200-250 Porto, Portugal; ccamiladias@gmail.com; 6CINTESIS–Center for Health Technology and Services Research, 4200-250 Porto, Portugal

**Keywords:** cardiovascular risk, cerebrovascular events, chronic kidney disease, endothelial dysfunction, reactive hyperemia index, peripheral arterial tonometry

## Abstract

Background: Patients with chronic kidney disease (CKD) have markedly increased rates of end stage renal disease, major adverse cardiovascular/cerebrovascular events (MACCEs), and mortality. Endothelial dysfunction (ED) is an early marker of atherosclerosis that is emerging as an increasingly important non-traditional cardiovascular risk factor in CKD. There is a lack of clinical studies examining the association between ED and both cardiovascular and renal endpoints in patients with CKD. Aims: We examined the association between reactive hyperemia index (RHI), a validated measure of endothelial function measured by peripheral arterial tonometry (PAT), with traditional cardiovascular risk factors in pre-dialysis CKD patients and prospectively evaluated the role of RHI as predictor of renal and cardiovascular outcomes in this population. Methods: One hundred and twenty pre-dialysis patients with CKD stages 1 to 5 (CKD group) and 18 healthy kidney donor candidates (control group) were recruited and had a successful RHI measurement by PAT. General demographic and clinical information including traditional cardiovascular risk factors were registered from all participants. Thereafter, patients were prospectively followed-up for a median time of 47 (IQR 19–66) months to determine associations of RHI with renal outcomes, MACCEs, hospitalizations or mortality. Results: In the CKD patient population, the mean age was 57.7 ± 15.5 years, the mean eGFR was 54.9 ± 36.7 mL/min/1.73 m^2^ (CKD-EPI) and 57 were males (47.5%). At baseline, in univariate analysis, RHI in the CKD group correlated positively with eGFR (r = 0.332, *p* < 0.0001) and correlated negatively with age (r = −0.469, *p* < 0.0001), Charlson index (r = −0.399, *p* < 0.0001), systolic blood pressure (r = −0.256, *p* = 0.005), and proteinuria (r = 0.211, *p* = 0.027). Reactive hyperemia index in the control group did not significantly differ from RHI observed in patients with CKD stages 1 to 5 (2.09 ± 0.40 vs. 2.01 ± 0.06, *p* = 0.493). In adjusted analysis, only age (β = −0.014, *p* = 0.003) remained independently associated with RHI at baseline. During follow-up, 8 patients suffered a MACCEs, 33 patients experienced renal function deterioration, 17 patients were hospitalized for medical reasons and 6 patients died. RHI at baseline was not significantly associated with CKD progression (1.94 vs. 2.02, *p* = 0.584), hospitalizations (1.90 vs. 2.04, *p* = 0.334), and all-cause mortality (1.65 vs. 2.01, *p* = 0.208) or MACCEs (1.77 vs. 2.01, *p* = 0.356), but was significantly associated with cerebrovascular events (1.27 vs. 2.02, *p* = 0.004) and with a composite cardiovascular outcome (MACCEs, hospital admissions and death; 1.73 vs. 2.07, *p* = 0.035). **Conclusion:** Our results suggest that RHI may be a predictor for the development of cerebrovascular events in pre-dialysis CKD patients who may benefit from more aggressive preventive measures.

## 1. Introduction

Chronic kidney disease (CKD) is one of the strongest risk factors for cardiovascular (CV) morbidity and mortality. Increased CV risk is observed even in the earlier stages of CKD and increases markedly as CKD progresses [1].

Indeed, a CKD patient is more likely to die from CV disease than to progress to end-stage renal disease [2]. Because diabetes mellitus and hypertension are the two main etiologies of CKD worldwide, the increased CV risk in this population was assumed for a long time to be the result of these two underlying conditions, which are simultaneously two traditional CV risk factors. However, meta-analyses clearly showed that impaired kidney function is a CV risk factor per se independently of both diabetes and hypertension, thus emphasizing the need to better understand the combined effects of a series of non-traditional CV risk factors in CKD [3,4].

Endothelial dysfunction (ED), one of the first steps in the development and progression of atherosclerosis, is considered a non-traditional CV risk factor in CKD and its prevalence progressively increases as the disease progresses to end stage kidney disease. ED is evident at an early stage in CKD patients, occurs independently of hypertension development and contributes to arterial stiffness and to renal interstitial fibrosis [5,6,7].

In addition to chronic inflammation and oxidative stress, hemodynamic disturbances, and the continuous exposure to uremic toxins in CKD, may result in an imbalance between endothelial injury and repair that progressively worsens as renal function deteriorates [8].

The assessment of ED was found to be useful to predict cardiovascular events, not only in the general population but also in high-risk groups [9,10]. Forearm reactive hyperemia has been considered a gold-standard measure of macrovascular endothelial function that proved to correlate well with invasively measured coronary ED [11,12]. However, this technique has been criticized for lack of reproducibility because it is highly operator dependent. Recently, reactive hyperemia index (RHI) derived from peripheral arterial tonometry (PAT) has gained wide acceptance in the scientific community because it is a non-invasive and non-operator dependent method for assessment of microvascular function, that has been validated for stratification in both low-risk and high-risk populations [13,14]. However, there are few studies that have examined the association between microvascular function assessed by PAT and CV risk factors in CKD patients and the results are conflicting. Moreover, it remains to be established if ED assessed by PAT may be useful as a reliable biomarker to predict CKD progression as well as major CV and cerebrovascular events (MACCEs) and mortality in this population. In the present study we examined the association between RHI measured by PAT with traditional cardiovascular risk factors in pre-dialysis CKD patients and prospectively evaluated the role of RHI as predictor of renal and CV outcomes in this population.

## 2. Methods

### 2.1. Study Population

We recruited pre-dialysis CKD patients, followed-up at the outpatient clinic of the Nephrology Dept. of São João University Hospital Center, Porto, Portugal. Patients were studied in standard conditions (no changes were made in their usual medication, nor was there extra reinforcement of diet restrictions or smoke abstinence). Patients with acute kidney injury, ongoing immunosuppression, recent hospital admission (<2 weeks), recent infections (<1 week), heart failure (diagnosed according to appropriate Framingham criteria) and known psychiatric disturbances were excluded from the study. The etiology of CKD was registered, and patients were distributed according to KDIGO CKD categories, using GFR estimated by CKD-EPI formula (stage 1: >90 mL/min/1.73 m^2^ and evidence of kidney damage; stage 2: between 60–89 mL/min/1.73 m^2^ and evidence of kidney damage; stage 3a: between 45–59 mL/min/1.73 m^2^ stage 3b: between 30–44 mL/min/1.73 m^2^; stage 4: between 15–29 mL/min/1.73 m^2^; stage 5: <15 mL/min/1.73 m^2^).

A control group of healthy subjects was also recruited. They were all kidney donor candidates followed-up at the pre-transplant outpatient clinic of São João University Hospital Center, Porto, Portugal. The healthy group was evaluated at hospital admission, the day before kidney donation.

### 2.2. Cross-Sectional Study

Anthropometric measurements, resting systolic and diastolic blood pressure (mean of the last 2 of 3 measurements), and a validated comorbidity index (Charlson Index) were assessed in both pre-dialysis CKD and control groups. Blood and urine samples were collected in all participants. Blood pressure was measured in the office at room temperature of 20–22 °C, with low light, no distractions, in the seated position, after at least 10 to 15 min of rest. Hypertension was diagnosed if systolic blood pressure (SBP) in the office was ≥140 mm Hg and/or diastolic blood pressure (DBP) was ≥90 mm Hg following repeated examination, or when patients presented controlled blood pressure values under anti-hypertensive pharmacologic treatment. Renal function (eGFR), proteinuria, phosphate (Pi), parathormone (iPTH), C reactive protein (CRP) and natriuretic peptide type B (BNP) serum levels were evaluated in all patients using standard laboratory methods. In addition, endothelial function was assessed by peripheral arterial tone (PAT) (Endo-Pat 2000, Itamar, Israel). This device allows non-invasive measurement of vasoreactivity without the disadvantages of conventional ultrasound measurement. The PAT serves as a measure of peripheral vasomotor function as was validated in the large, community-based Framingham study [15]. The EndoPAT detects plethysmographic pressure changes in the fingertips caused by the arterial pulse and translates this to PAT. Endothelium-mediated changes in vascular tone after occlusion of the brachial artery reflect a downstream hyperemic response, which is a measure for arterial endothelial function. Measurements on the contralateral arm are used to control for concurrent non endothelium-dependent changes in vascular tone. The technique provides values for the calculation of RHI, which gives an indication of the endothelial vasodilator function. RHI is the post-to-pre occlusion PAT signal ratio in the occluded arm, relative to the same ratio in the control arm, and corrected for baseline vascular tone [16]. Lower RHI values correspond to greater ED. RHI below 1.65 is considered severe ED, whereas RHI between 1.66 and 1.99 is considered moderate endothelial dysfunction.

### 2.3. Prospective Study

All recruited CKD patients were prospectively followed-up for a median of 47 (IQR 19–66) months, to evaluate hard renal and CV outcomes including progression of CKD and ESRD, hospitalizations, MACCEs and CV/all-cause mortality. The MACCEs included acute coronary syndrome (ACS), heart failure, and stroke.

Acute coronary syndrome (ACS) was defined as a clinical diagnosis of ST-segment elevation myocardial infarction (STEMI), non-STEMI, or unstable angina pectoris. Heart failure was diagnosed when the patient met the appropriate Framingham criteria [17]. Cardiovascular death was defined as mortality due to a heart-related cause (death attributable to ACS, heart failure, arrhythmia, or sudden death) or to a cerebrovascular event. Hospitalizations included non-programmed, more than 24 h hospital admissions for medical reasons. Admissions for trauma, surgery or other scheduled procedures were not considered. A composite cardiovascular outcome was established including MACCEs, hospital admission, and deaths. Renal outcomes included CKD progression, defined as serum creatinine doubling or a >50% decrease in eGFR according to CKD-EPI formula or renal replacement therapy initiation (ESRD) after enrollment.

### 2.4. Statistical Analysis

Categorical variables were described through absolute (n) and relative (%) frequencies, while continuous variables were described as mean and standard deviation, or median, interquartile (IQR) range, and minimum and maximum, when appropriate. Differences in continuous variables were assessed by Mann–Whitney *U* test or T test for independent samples, while chi-square tests were used to analyze differences in categorical variables. A Dunnet post hoc test was used to compare different CKD stages and the control group. A correlation analysis was performed using Spearman correlation coefficients. Linear regression was applied to determine the relationship between RHI and renal function adjusted for possible confounders. Coefficient regression (beta) and 95% confidence intervals (95% CI) are presented. All reported *p*-values were two-sided, and the significance level was set at 5%. All analyses were conducted using SPSS software (Version 26.0 for Windows, SPSS, Chicago, IL, USA).

### 2.5. Ethics

The research was approved by the Ethics Committee for Health and the Local Institutional Review Board of São João University Hospital Centre and was carried out in accordance with the Declaration of Helsinki (2008) of the World Medical Association.

## 3. Results

### 3.1. Cross-Sectional Study

One hundred and twenty patients (57 males, 47.5%; mean age 57.7 ± 15.5 years) and 18 healthy volunteers (2 males, 11%; mean age 48.8 ± 8.4 years), were enrolled in the study.

The pre-dialysis population presented a higher mean body mass index (BMI): 28.4 ± 5.9 vs. 25.3 ± 2.8 kg/m^2^ (*p* = 0.006), a higher Charlson index: 3.9 ± 2.9 vs. 0.6 ± 0.6 (*p* < 0.001) and as expected, a lower eGFR: 54.9 ± 36.7 vs. 111.9 ± 11.2 mL/min/1.73 m^2^ (*p* < 0.001) compared with the control group. The leading etiologies of CKD in the pre-dialysis patient group were undetermined (13%) and diabetic nephropathy (11.6%). Hypertension, dyslipidemia, diabetes and cardiovascular disease were present in 80%, 57%, 30%, and 27% of the CKD patients, respectively. In the CKD group, 48% and 67% of the patients were treated with ACEi and statins respectively. The characterization of the control group and the pre-dialysis CKD population according to KDIGO stages is presented in Table 1. Reactive hyperemia index in the control group did not significantly differ from RHI observed in patients with CKD stages 1 to 5 (2.09 ± 0.40 vs. 2.01 ± 0.06, *p* = 0.493). However, in the pre-dialysis CKD group RHI was significantly lower in patients with CKD stages 4 (1.71 ± 0.56, *p* = 0.003) and 5 (1.67 ± 0.38, *p* = 0.033), in comparison with RHI observed in patients included in CKD stage 1 (Table 1) i.e., the patients included in the more advanced CKD stages presented worse vascular reactivity. In line with this data, RHI proved to be positively correlated with eGFR (r = 0.332, *p* < 0.001) (Table 2; Figure 1). RHI was also positively correlated with HDL cholesterol (r = 0.190, *p* = 0.043). In addition, RHI was negatively correlated with age (r = −0.469, *p* < 0.001), Charlson index (r = −0.399, *p* < 0.001), systolic blood pressure (r = −0.256, *p* = 0.005), proteinuria (r = 0.211, *p* = 0.027), triglycerides (r = 0.255, *p* = 0.006), parathormone (r = 0.283, *p* = 0.004), and BNP (r = −0.407, *p* = 0.001) (Table 2). Pre-dialysis CKD patients with hypertension (1.88 vs. 2.53, *p* = 0.002), diabetes (1.76 vs. 2.12, *p* = 0.001), dyslipidemia (1.88 vs. 2.02, *p* = 0.018), cardiovascular (1.82 vs. 2.08, *p* = 0.029), cerebrovascular disease (1.63 vs. 2.06, *p* = 0.036) as well as those treated with statins (1.84 vs. 2.23, *p* = 0.002) presented significantly lower RHI (Table 3).

The above-mentioned comorbidities were included in a stepwise regression analysis and eGFR (β = 0.004, *p* = 0.047) and hypertension (β = −0.390, *p* = 0.023) remained independent predictors for RHI in this setting. However, when age was included in the analysis, this relation was no longer observed (Table 4).

### 3.2. Prospective Study

The pre-dialysis CKD population was prospectively followed up for a median time of 47 (IQR 19–66) months. During follow-up, 8 patients suffered a MACCE (3 patients an acute myocardial infarction, 4 patients a cerebrovascular event, 1 patient a mesenteric ischemia), 33 patients experienced renal function deterioration, of which 17 started dialysis. Seventeen patients were hospitalized for medical reasons and 6 patients died. In 3 out of the 6 patients who died, a MACCE was the main cause of death. Twenty-one patients reached the combined cardiovascular outcome (Table 5). We then explored further the association of RHI and renal, cardiovascular outcomes, hospitalizations, and overall survival. Reactive hyperemia index was not significantly associated with CKD progression (1.94 vs. 2.02, *p* = 0.584), ESRD (1.77 vs. 2.05, *p* = 0.120), hospitalizations (1.90 vs. 2.04, *p* = 0.334), all-cause mortality (1.65 vs. 2.01, *p* = 0.208) or MACCEs (1.77 vs. 2.01, *p* = 0.356). However, RHI was significantly associated with cerebrovascular events (1.27 vs. 2.02, *p* = 0.004) as well as with the composite cardiovascular outcome (MACCEs, hospital admissions, and death: 1.73 vs. 2.07, *p* = 0.035) (Table 6). None of the patients who had a cerebrovascular event during follow-up had previously known cerebrovascular disease; however, they all had hypertension, and moderate to severe CKD.

## 4. Discussion

The present study assessed endothelial function by RHI-PAT in a non-dialysis CKD population and in healthy subjects and explored the association of RHI with general CV risk factors as well as the predictive role of RHI regarding renal and cardiovascular outcomes in the CKD population, prospectively followed-up for approximately four years. The main findings were the following: (i) Although RHI was significantly lower in patients with CKD stages 4 and 5 compared with patients with CKD stages 1, in healthy subjects RHI did not differ significantly from the observed in non-dialysis CKD population; (ii) using multivariate logistic regression analysis, RHI-PAT was not related with most general CV risk factors in CKD patients, although a significant association was found with both eGFR and hypertension when age was not included in the analysis; (iii) during a mean follow-up period of 47 months of the non-dialysis CKD population, RHI-PAT at baseline was a predictor of cerebrovascular events as well as of the composite of CV outcome. Taken together, our results indicate that ED, non-invasively assessed by RHI-PAT, may provide information for CV risk stratification in this population, identifying patients vulnerable to cerebrovascular events during follow-up.

Chronic kidney disease is associated with abnormalities of vascular function (including ED) that have been implicated in the disproportionately high CV morbidity and mortality in this population. A number of disturbances have been suggested to contribute to ED in CKD including elevated blood pressure, insulin resistance, abnormalities in the nitric oxide pathway (ADMA and L-arginine), inflammation (sVCAM-1 and sE-selectin), thrombosis (vWF), and endothelial dependent dilation (FMD) [18,19]. However, the direct effect of the decrease in eGFR on endothelial function and the associated CV risk is far from being established. In this study, we found that in univariate analysis patients with advanced CKD presented a reduced endothelial function compared to the those in early stages of the disease, and this reduction correlated with the severity of renal failure. Also, in univariate analysis RHI correlated with CV risk factors in our pre-dialysis CKD population. Nevertheless, in age adjusted analysis there was lack of significant association of lower RHI with both reduced eGFR and CV risk factors. These results agree with the observations of Wang et al., that found that RHI did not decline with reduced renal function in non-dialysis CKD patients and had a modest association with traditional CV risk factors [20]. These results using RHI-PAT differ from those that used other methods for the assessment of endothelial function in CKD patients, namely flow-mediated vasodilation (FMD), that reported a high prevalence of ED in this population which correlated well with the decrease of renal function [21,22].

The lack of association of RHI-PAT with both CV risk factors and eGFR might be related to the role played by nitric oxide in the regulation of the digital hyperemia response. Actually, evidence has been gathered that the inhibition of nitric oxide synthesis completely abolishes FMD whereas it only reduces by 50% the digital hyperemia response [23,24]. One can thus hypothesize that the endothelial-independent mechanisms underlying the digital hyperemia response may be more influenced by factors yet to be identified than general CV risk factors including the decrease in eGFR. According to this view, we found in the present study that RHI in healthy subjects did not differ significantly from that observed in the CKD group. Our results agree well with those from Moerland et al., that compared RHI between 12 healthy subjects and 6 CKD patients and found that RHI did not significantly differ between the two groups [25].

In the present study, age proved to be the independent predictor of RHI in the stepwise regression analysis. Our results fit well with the findings from the Brazilian Longitudinal Study of Adult Health cohort study that reported a negative association between age and RHI [26].

We must emphasize, however, that older patients are highly prevalent in advanced CKD stages and are more prone to hypertension, which can underestimate the real influence of renal dysfunction and blood pressure on RHI. A previous study evaluated the effect of aging on microvascular reactivity in patients with advanced CKD [27]. The authors demonstrated that microvascular reactivity in young patients with advanced CKD was identical to old healthy controls and old patients with CKD. Particularly in the elderly, the uremic state itself seems to have less additional negative burden in an already compromised microvascular reactivity induced by aging [27].

The lack of association of RHI with general CV risk factors and eGFR in age-adjusted analysis in our study does not exclude the possibility that RHI may be useful to predict CV outcomes in these patients as was found in other populations [13,28,29], as it has been suggested that impaired RHI may represent a novel pathophysiologic pathway to CV events [20].

Hirata et al., demonstrated that peripheral ED assessed by RHI correlated significantly with the presence of coronary artery disease in stable CKD patients and independently predicted cardiovascular event risk in these CKD patients [30]. It should be noted, however, that the CKD population studied by Hirata et al., was chosen from a group of patients with suspected coronary artery disease which may be not representative of the more general CKD population [30]. In the present study we found in our general non-dialysis CKD population prospectively followed-up for approximately 4 years that RHI at baseline predicted cerebrovascular events and was also associated with a composite CV outcome during follow-up. None of the patients who had experienced a cerebrovascular event had previously known cerebrovascular disease, but they were all hypertensive and had moderate to severe CKD. It has been described in the general population that ED significantly predicts cerebrovascular events independently of traditional risk factors [31]. In fact, Santos-Garcia et al. calculated a brachial artery FMD cut-off value inferior or equal to 4.5% as an independent predictor of new-onset vascular events, including stroke recurrence after the acute phase of ischemic stroke [32]. The prevalence of cerebrovascular events is higher in CKD patients than that in the general population [33]. However, there is conflicting epidemiological data about whether low eGFR may be a risk factor for stroke independent of traditional cardiovascular risk factors that are common to both entities [34,35]. Novel non-traditional risk factors are also associated with CKD, namely chronic inflammation, oxidative stress, asymmetric dimethylarginine, sympathetic nerve overactivity, thrombogenic factors, and hyperhomocysteinaemia, which may also contribute to the excess risk of cerebrovascular disease in patients with CKD by triggering vascular injury and ED [33,36]. We can hypothesize that renal impairment associated with other CV risk factors namely hypertension, induces ED and potentiates the risk of the development of stroke in the pre-dialysis stages. In line with these findings, we would expect an association between ED and CKD progression, ESRD, hospitalizations, all-cause mortality, and MACCEs, however, despite the high number of CV risk factors observed in our population, the number of events observed in the prospective study was relatively small and this fact may justify the lack of association between ED, renal, and other CV outcomes besides cerebrovascular events and the composite cardiovascular outcome.

We acknowledge some limitations of our study. First, this was a single center observational study with a small number of patients and events, so we cannot demonstrate a direct cause-and-effect risk association. Confirmation by larger prospective investigations is required. Second, RHI was measured at a single time point at baseline and so we could not capture changes in RHI with the decrease of renal function. Third, the fact that CKD groups had different mean ages prevents us from unequivocally assuming that endothelial dysfunction decline is dependent of renal function deterioration. Nevertheless, we also emphasize that epidemiologically older patients are more prevalent in advanced CKD stages, and this is a hard limitation to overcome. Our study also has some strengths. To our knowledge this is the first study that evaluated prospectively the role of RHI-PAT as a predictor of renal and cardiovascular outcomes in a general non-dialysis CKD patient population. We were also able to compare RHI between patients with CKD and healthy controls.

In conclusion, our results demonstrate that RHI, measured by PAT, declined with renal function but this association is age-dependent and there was no difference between the general non-dialysis CKD population and healthy subjects. Our findings also suggest that RHI may be a predictor for the development of cerebrovascular events in pre-dialysis CKD patients who may thus benefit from more aggressive preventive measures.

## Figures and Tables

**Figure 1 life-11-00128-f001:**
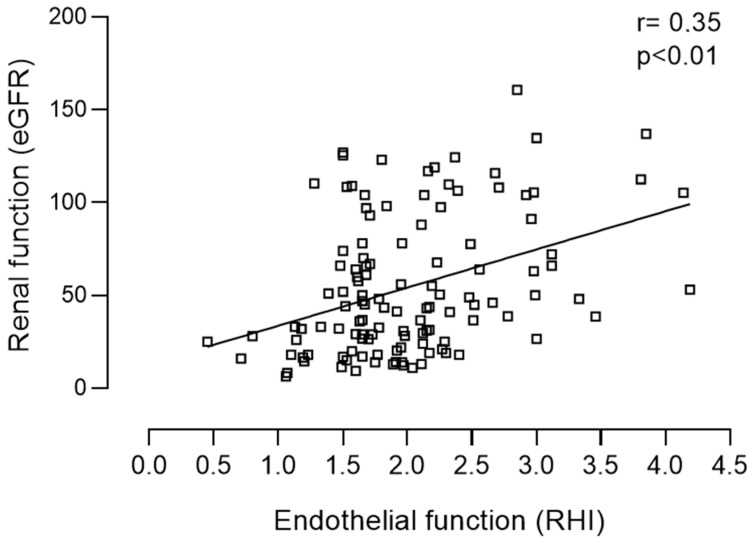
Positive correlation between renal (eGFR) and endothelial function (RHI) in the pre-dialysis population.

**Table 1 life-11-00128-t001:** Characterization of the pre-dialysis patient population by CKD stages and the control group.

	Control (n = 18)	Stage 1 (n = 27)	Stage 2 (n = 16)	Stage 3a (n = 14)	Stage 3b (n = 22)	Stage 4 (n = 29)	Stage 5 (n = 12)
DEMOGRAPHIC DATA							
Age (years), mean ± sd	48.8 ± 8.4	46.5 ± 11.7	53.7 ± 14.8	56.6 ± 17.1	61.0 ± 14.7	65.4 ± 15.2 (*p* = 0.004) *	65.2 ± 7.3 (*p* = 0.026) *
Gender (n, %)	2 (11.1)	9 (33.3)	8 (50.0)	8 (57.4)	14 (63.6)	13 (44.8)	5 (41.7)
Body mass index mean ± sd	25.3 ± 2.8	27.7 ± 4.2	30.1 ± 7.6	29.0 ± 4.7	30.2 ± 6.7	25.3 ± 4.8	30.8 ± 6.5
Diabetes, n (%)	0 (0)	1 (4)	5 (31)	3 (21)	8 (36)	12 (41)	7 (58)
Hypertension n (%)	1 (7)	13 (48)	13 (81)	10 (71)	21 (95)	27 (93)	12 (100)
Systolic blood pressure, (mmHg)	125 ± 14	124 ± 15	134 ± 18	123 ± 13	139 ± 27	138 ± 21	133 ± 9
Diastolic blood pressure (mmHg)	74 ± 8	75 ± 13	75 ± 13	79 ± 11	76 ± 14	74 ± 13	76 ± 13
Heart rate (bpm)	71 ± 10	71 ± 14	73 ± 13	66 ± 13	68 ± 13	73 ± 12	73 ± 13
Cardiovascular disease, n (%)	0 (0)	0 (0)	5 (31)	3 (21)	10 (45)	9 (31)	5 (42)
CKD RELATED PARAMETERS							
eGFR CKD-EPI (ml/min/1.73 m^2^) mean ± sd	111.9 ±11.2	112.8 ± 2.9	70.1 ± 1.8	51.7 ± 1.1	38.3 ± 1.1	22.8 ± 0.9	12.1 ± 0.7
Calcium (mg/dL) mean ± sd	4.63 ± 0.18	4.72 ± 0.24	4.69 ± 0.29	4.76 ± 0.36	4.80 ± 0.28	4.62 ± 0.26	4.68 ± 0.33
Phosphate (mg/dL) mean ± sd	3.28 ± 0.47	3.01 ± 0.37	2.96 ± 0.45	3.36 ± 0.51	3.47 ± 0.70	3.90 ± 0.61 (*p* = 0.003) *	4.09 ± 0.54 (*p* = 0.001) *
Parathormone (pg/mL) mean ± sd	45.4 ± 14.28	43.0 ± 16.4	70.3 ± 31.8	63.5 ± 23.1	86.0 ± 44.7	166.7 ± 154.3 (*p* < 0.001) *	179.6 ± 96.2 (*p* = 0.001) *
Protein/creatinine ratio (mg/g), median (IQR)	82.3 (65.0–117.0)	252.0 (80.0–669.5)	170.1 (85.3–780.1)	462.0 (185.0–1035.4)	329.2 (151.7–863.3)	888.7 (362.0–1737.0)	1199.0 (206.0–3975.0) (*p* = 0.020) *
ENDOTHELIAL FUNCTION, RHI score, mean ± sd	2.09 ± 0.40	2.36 ± 0.77 ^†,‡^	2.10 ± 0.59	2.14 ± 0.76	2.03 ± 0.62	1.71 ± 0.56 ^†^ (*p* = 0.003)	1.67 ± 0.38 ^‡^ (*p* = 0.033)
CARDIOVASCULAR RELATED PARAMETERS							
Charlson Index score mean ± sd	0.6 ± 0.6	0.7 ± 0.9	2.6 ± 2.3 (*p* = 0.043) *	4.1 ± 3.1 (*p* < 0.001) *	4.8 ± 2.3 (*p* < 0.001) *	5.8 ± 2.3 (*p* < 0.001) *	6.0 ± 2.2 (*p* < 0.001) *
Sedimentation velocity mean ± sd	14 ± 13	27 ± 22	33 ± 25	28 ± 25	35 ± 23	54 ± 33 (*p* < 0.001) *	58 ± 31 (*p* = 0.018) *
Albumin (g/dL) mean ± sd	43.1 ± 2.6	41.7 ± 3.4	39.5 ± 4.1 (*p* = 0.010) *	42.1 ± 4.9	39.1 ± 8.4	38.8 ± 4 (*p* = 0.050) *	38.7 ± 3.6
Total Cholesterol (mg/dL) mean ± sd	174 ± 31	196 ± 36	179 ± 50	194 ± 36	172 ± 36	174 ± 40	198 ± 63
HDL Cholesterol (mg/dL) mean ± sd	59 ± 14	55 ± 14	52 ± 14	53 ± 13	51 ± 15	48 ± 13	46 ± 18
Triglycerides (mg/dL) mean ± sd	82 ± 25	127 ± 97	142 ± 91	121 ± 37	121 ± 53	165 ± 82	341 ± 559 (*p* = 0.002) *
Acid uric mean ± sd	4.1 ± 0.9	5.2 ± 1.6	6.3 ± 1.9 (*p* = 0.004) *	6.8 ± 2.0 *p* < 0.001) *	6.7 ± 1.3 (*p* < 0.001) *	7.6 ± 1.5 (*p* < 0.001) *	7.7 ± 2.5 (*p* < 0.001) *
C reactive protein (mg/L) median (IQR)	1.2 (0.4–2.8)	2.0 (1.1–5.9)	2.8 (1.1–9.6)	3.3 (1.7–6.1)	2.7 (1.3–6.1)	1.9 (1.1–5.5)	3.1 (1.1–23.0)
BNP (pg/mL) median (IQR)	19.1 (10.0–30.2)	26.7 (15.0–34.9)	27.0 (17.0–69.5)	84.0 (67.0–105.0)	94.5 (74.8–159.2)	145.5 (49.2–339.4) (*p* = 0.010) *	46.0 (31.2–263.0) (*p* = 0.004) *
Left Ventricular Mass (g) mean ± sd	139.6 ± 24.9	182.4 ± 35.2	271.9 ± 149.2	165.4 ± 73.9	217.9 ± 72.9	207.6 ± 56.7	212.7 ± 81.5
Ejection Fraction (%) mean ± sd	67 ± 5	60 ± 4	62 ± 9	63 ± 10	60 ± 6	56 ± 3	54 ± 24

* The *p*-value presented was the comparison between stages and control group using Dunnet post hoc test. ^†^ Comparison between stage 1 vs. 4. ^‡^ Comparison between stage 1 vs. 5.

**Table 2 life-11-00128-t002:** Correlation between endothelial function (RHI) and clinical variables in pre-dialysis CKD cohort of patients (n = 120).

	RHI Correlation Coefficient
Age (y)	−0.469 **
Charlson Index	−0.399 **
Body mass index (kg/m^2^)	−0.174
Systolic blood pressure (mmHg)	−0.256 **
Diastolic blood pressure (mmHg)	−0.087
Heart rate (bpm)	−0.003
eGFR CKD-EPI (ml/min/1.73 m^2^)	0.348 **
Protein/creatinine ratio (mg/g)	−0.211 *
Reactive protein-C (mg/L)	−0.061
HDL Cholesterol (mg/dL)	0.190 *
LDL Cholesterol (mg/dL)	0.051
Triglycerides (mg/dL)	−0.255 **
Alkaline Phosphatase (mg/dL)	−0.136
Parathormone (pg/mL)	−0.283 **
BNP (pg/mL)	−0.407 **
Left ventricular mass (g)	−0.139
Ejection Fraction (%)	0.147

* *p* < 0.05; ** *p* < 0.01.

**Table 3 life-11-00128-t003:** Association between endothelial function (RHI) and cardiovascular risk factors in the pre-dialysis CKD cohort of patients (n = 120).

		Reactive Hyperemia Index (RHI)		
		Mean	Standard Deviation	*p* Value ^1^
Gender	Female	2.10	0.72	0.113
	Male	1.91	0.61	
Hypertension	No	2.53	0.87	0.002
	Yes	1.88	0.55	
Diabetes	No	2.12	0.73	0.001
	Yes	1.76	0.45	
Dyslipidemia	No	2.19	0.75	0.018
	Yes	1.88	0.59	
Cardiovascular disease	No	2.08	0.72	0.029
	Yes	1.82	0.49	
Cerebrovascular disease	No	2.06	0.68	0.036
	Yes	1.63	0.48	
Statins	No	2.23	0.73	0.002
	Yes	1.84	0.58	
ACEi	No	2.04	0.71	0.527
	Yes	1.96	0.62	

^1^*t* test for independent sample.

**Table 4 life-11-00128-t004:** Linear regression: association between RHI (dependent variable) and eGFR, adjusted to possible confounders.

	Beta	95% CI		*p*-Value
Model 1				
Renal function (eGFR-CKD Epi)	0.007	0.004	0.010	<0.001
Model 2				
Renal function (eGFR-CKD Epi)	0.005	0.001	0.008	0.010
Hypertension	−0.441	−0.758	−0.124	0.007
Model 3				
Renal function (eGFR-CKD Epi)	0.006	0.003	0.009	<0.001
Diabetes	−0.192	−0.456	0.071	0.152
Model 4				
Renal function (eGFR-CKD Epi)	0.006	0.003	0.009	<0.001
Dyslipidemia	−0.175	−0.415	0.064	0.150
Model 5				
Renal function (eGFR-CKD Epi)	0.006	0.003	0.010	<0.001
Cerebrovascular disease	−0.251	−0.639	0.136	0.202
Model 6				
Renal function (eGFR-CKD Epi)	0.007	0.003	0.010	<0.001
Cardiovascular disease	−0.085	−0.356	−187	0.538
Model 7				
Renal function (eGFR-CKD Epi)	0.003	0.000	0.006	0.086
Age	−0.018	−0.026	−0.010	<0.001
Model 8				
Renal function (eGFR-CKD Epi)	0.007	0.004	0.010	<0.001
Gender (male)	−0.128	−0.356	0.101	0.272
Model 9				
Renal function (eGFR-CKD Epi)	0.005	0.002	0.009	0.001
Body Mass Index	−0.017	−0.036	−0.003	0.096
Model 10				
Renal function (eGFR-CKD Epi)	0.006	0.003	0.009	0.001
Systolic Blood Pressure	−0.006	−0.011	0.000	0.041
Model 11				
Renal function (eGFR-CKD Epi)	0.007	0.004	0.010	<0.001
Albuminuria	−0.008	−0.031	0.014	0.460
Model 12				
Renal function (eGFR-CKD Epi)	0.004	0.000	0.007	0.047
Hypertension	−0.390	−0.725	−0.055	0.023
Diabetes	−0.164	−0.459	0.130	0.271
Dyslipidemia	−0.075	−0.328	0.178	0.557
Cerebrovascular disease	−0.228	−0.625	0.169	0.258
Cardiovascular disease	0.084	−0.221	0.388	0.587
Model 12				
Renal function (eGFR-CKD Epi)	0.002	−0.002	0.006	0.367
Hypertension	−0.165	−0.509	0.180	0.345
Diabetes	−0.031	−0.339	0.276	0.840
Dyslipidemia	0.040	−0.346	0.425	0.838
Cerebrovascular disease	0.198	−0.129	0.525	0.232
Cardiovascular disease	−0.015	−0.443	0.413	0.945
Age	−0.014	−0.023	−0.005	0.003
Gender	−0.085	−0.339	0.170	0.510
Body Mass Index	−0.017	−0.038	0.004	0.119
Albuminuria	−0.004	−0.033	0.026	0.812

**Table 5 life-11-00128-t005:** Cardiovascular and renal outcomes during follow-up in the studied population.

FOLLOW-UP (Months)	47 (19–66)
**Outcome**	n (%)
MACCEs	8 (7.9)
*Acute myocardial infarction*	*3 (2.97)*
*Stroke*	*4 (3.96)*
Composite outcome on CKD progression	33 (33.0)
*Progression to ESRD*	17 (17.0)
Hospital admission for medical causes	17 (16.8)
Death	6 (5.9)
*Death by MACCEs*	3 (3.0)
Composite cardiovascular outcome *	21 (17.5)

MACCEs: major adverse cardiovascular and cerebrovascular events; ESRD: End Stage Renal Disease. * composite cardiovascular outcome: MACCEs or Hospital admission for medical reasons or deaths.

**Table 6 life-11-00128-t006:** Associations between RHI and renal, cardiovascular and survival outcomes during a 4-year follow-up in the pre-dialysis population (n = 120).

		Reactive Hyperemia Index (RHI)		
		Mean	sd	*p*-Value ^1^
**1.MACCEs**	No	2.01	0.68	0.356
	Yes	1.77	0.68	
1.1. Acute myocardial infarction	No	1.99	0.69	0.957
	Yes	2.01	0.44	
**1.2. Cerebrovascular events**	**No**	**2.02**	**0.68**	**0.004**
	**Yes**	**1.27**	**0.26**	
2.Death	No	2.01	0.69	0.208
	Yes	1.65	0.58	
3.CKD progression	No	2.02	0.71	0.584
	Yes	1.94	0.62	
3.1. Progression to ESRD	No	2.05	0.70	0.120
	Yes	1.77	0.59	
4.Hospital admission	No	2.04	0.73	0.334
	Yes	1.90	0.59	
**5.Composite cardiovascular outcome ***	**No**	**2.07**	**0.69**	**0.035**
	**Yes**	**1.73**	**0.53**	

^1^*t* test for independent sample. * composite outcome: MACCEs or Hospital admission for medical reasons or deaths.

## Data Availability

The data presented in this study are available on request from the corresponding author.

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
