# Peer review of "Endothelial Dysfunction Is Associated with Cerebrovascular Events in Pre-Dialysis CKD Patients: A Prospective Study"

_life, 2021, doi:10.3390/life11020128_

Round 1
Reviewer 1 Report
This is a very interesting paper providing rationale to use RHI assessed by PAT to predict the development of cerebrovascular events in CKD patients.
In my opinion, the following corrections could improve the quality of this paper:
- The information on drugs used at baseline (at least these known to influence endothelial dysfunction, such as statins or ACEi) is lacking. This information would help understanding major findings, particularly in the context of significant RHI correlations with HDL and triglycerides or with the hypertension.
- Authors found that RHI was negatively correlated with age, whereas patients in CKD 4 and 5 were significantly older than the remaining subjects. In the methods section it is explained that the relationship between RHI and eGFR is adjusted for comorbidities. Age adjustment should be also included and discussed.
Reviewer 2 Report
- Methods: provide more data on the validity of the technique of PAT used to assess endothelial function in this cohort. Which is the agreement with the reference-standard technique of FMD? Which is the short-term reproducibility of this technique?
- Methods: provide more data on the technique of BP measurement at baseline.
- Methods: describe the adjudication of cardiovascular and renal events over the follow-up.
- Statistical analysis: provide a sample size calculation for this study.
- Table 1: provide a p value for the overall trend across groups. The reported p values for separate between-group comparisons should be corrected for multiple hypothesis testing. Significant difference may simply reflect false discovery due to the multiple t-tests.
- Table 2.3: the final fully adjusted model includes only adjustment for comorbidities. This model should be further adjusted for several other confounding factors (i.e. age, gender, BMI, SBP, albuminuria, etch).
- Table 3: the overall number of events is too small. The authors should define 1 combined cardiovascular outcome (i.e., all-cause death, non-fatal MI, non-fatal stroke, hospitalization due to CHF) and a combined kidney outcome (i.e., doubling of serum creatinine or initiation of dialysis).
- Table 4 reports only subgroup comparisons. There was nowhere a comprehensive survival analysis using univariate and multivariate Cox-regression models as well as Kaplan-Myer curves.
- Discussion: the study has several limitations that need to be adequately acknowledged. Most importantly, the study follows an observational design and can not demonstrate direct cause-and-effect risk associations.
Round 2
Reviewer 2 Report
No further comments